# Risk Factors for Spine Reoperation and Joint Replacement Surgeries after Short-Segment Lumbar Spinal Surgeries for Lumbar Degenerative Disc Disease: A Population-Based Cohort Study

**DOI:** 10.3390/jcm10215138

**Published:** 2021-10-31

**Authors:** Meng-Huang Wu, Christopher Wu, Jiann-Her Lin, Li-Ying Chen, Ching-Yu Lee, Tsung-Jen Huang, Yi-Chen Hsieh, Li-Nien Chien

**Affiliations:** 1Department of Orthopedics, Taipei Medical University Hospital, Taipei 110301, Taiwan; maxwutmu@gmail.com (M.-H.W.); lee161022@tmu.edu.tw (C.-Y.L.); tjdhuang@tmu.edu.tw (T.-J.H.); 2Department of Orthopaedics, School of Medicine, College of Medicine Taipei Medical University, Taipei 110301, Taiwan; 3College of Medicine, Taipei Medical University, Taipei 110301, Taiwan; cjwuchris@gmail.com; 4Division of Neurosurgery, Department of Surgery, Taipei Medical University Hospital, Taipei 110301, Taiwan; jiannher@tmu.edu.tw; 5Health Data Analytics and Statistics Center, Office of Data Science, Taipei Medical University, Taipei 110301, Taiwan; lychen@tmu.edu.tw; 6PhD Program of Neural Regenerative Medicine, College of Medical Science and Technology, Taipei Medical University, Taipei 110301, Taiwan; 7School of Health Care Administration, College of Management, Taipei Medical University, Taipei 110301, Taiwan

**Keywords:** disc degeneration disease, lumbar short-segment spinal surgery, reoperation, total joint replacement, fusion

## Abstract

Background: Short-segment lumbar spinal surgery is the most performed procedure for treatment of degenerative disc disease. However, population-based data regarding reoperation and joint replacement surgeries after short-segment lumbar spinal surgery is limited. Methods: The study was a retrospective cohort design using the Taiwan National Health Insurance Research Database for data collection. Patients selected were diagnosed with lumbar degenerative disc disease and undergone lumbar discectomy surgery between 2002 and 2013. The Kaplan–Meier method was used to estimate the incidence of 1-year spine reoperation and joint replacement surgeries, and the Cox proportional hazard regression was used to examine risk factors associated with the outcomes of interest. Results: A total of 90,105 patients were included. Incidences of 1-year spine reoperation and joint replacement surgeries for the hip and knee were 0.27, 0.04, and 0.04 per 100 people/month. Compared to fusion with the fixation group, fusion without fixation and the non-fusion group had higher risks of spine reoperation. Risk factors associated with spine reoperation included fusion without fixation, non-fusion surgery, age ≥ 45 years old, male gender, diabetes, a Charlson Comorbidity Index = 0, lowest social economic status, and steroid use history. Spine surgeries were not risk factors for joint replacement surgeries. Conclusions: Non-fusion surgery and spinal fusion without fixation had higher risks for spine reoperation. Spine surgeries did not increase the risk for joint replacement surgeries.

## 1. Introduction

Lumbar degenerative disc disease (DDD) is a common spine pathology causing disability [1]. Treatment for painful DDD focuses on minimizing pain, stabilizing the spine, and improving or maintaining mobility. DDD can usually be treated with a combination of pain management techniques and physical therapy. For a small percentage of patients, surgery may become an alternative option when conservative treatment has not worked well and their severe pain as well as muscle spasms make it difficult to function normally [1].

Short-segment lumbar spinal surgery is the most performed procedure for treatment of radiculopathy caused by DDD. Compared to decompression alone, spine fusion is normally viewed as a stabilizing treatment that may reduce the need for additional surgery. However, indications for fusion surgery in degenerative spine disorders remain controversial and the effects of fusion on spine reoperation rates are unclear [2]. Repeat lumbar spine operations are generally undesirable, implying persistent symptoms, progression of degenerative changes, or treatment complications. Additionally, persistent hip and knee joint pain after spine surgery also lead to sequential total joint surgeries, which have been reported as hip–spine syndrome and knee–spine syndrome [3,4]. Degenerative changes of the knee often cause loss of extension [5]. This may affect aspects of posture such as lumbar lordosis and lead to spine degeneration. These small groups of patients often have worse outcomes due to the coexisting joint problems.

Due to the increased use of spine surgery and the strain on health expenses, settlements from hospitals linked to complications and spine reoperation rates are needed to address the predictive influences for these events [6,7]. However, population-based data regarding spine reoperation and joint replacement surgeries after short-segment lumbar spinal surgery is limited. Therefore, the present study aims to investigate the incidence and risk factors for spine reoperation and joint replacement surgeries, including total hip replacement (THR) and total knee replacement (TKR), after short-segment lumbar spinal surgery for DDD.

## 2. Materials and Methods

### 2.1. Data Sources

The study was a retrospective cohort design and we used the Taiwan National Health Insurance Research Database (NHIRD) for data collection. NHIRD is a claim-based database including almost all health service claims of the beneficiaries enrolled in the National Health Insurance (NHI) in Taiwan. More than 99% of the population is enrolled in the mandatory, single-payer NHI program that provides comprehensive care, including outpatient and inpatient services, laboratory tests, and prescription drugs. Additionally, we also used the National Death Registry to obtain death records. All datasets can be linked by encrypted identifiers and confidentiality was ensured by abiding to data regulations of the Health and Welfare Data Science Center (HWDC), Ministry of Health and Welfare, Executive Yuan, Taiwan.

### 2.2. Study Samples

Patients diagnosed with lumbar DDD and who initially received lumbar discectomy surgery between 2002 and 2013 were included. The admission date of lumbar discectomy surgery was treated as the index date (referred to as index date hereafter). Regarding patients, we excluded those who (1) were younger than 18 years old, had sex information missing, or was not a citizen in Taiwan; (2) received THR or TKR before the index date; (3) had a diagnosis of a malignant tumor or traumatic injury; and (4) had pathological fracture, vertebral fracture, or a surgical procedure involving in ≥4 spinal vertebrae or pedicle screw implantations >4 levels. The exclusion was made to ensure the patient received short-segment lumbar spinal surgeries for lumbar DDD. The final lumbar discectomy surgery cohort was classified into three surgical groups including fusion with fixation, fusion without fixation, and non-fusion based on the procedures received during admission. The patient selection process is presented in Figure 1.

### 2.3. Study Outcomes

Information of the outcomes of interest were retrieved from the NHIRD by identifying the current procedural terminology surgery codes. The spine reoperation was recorded if the patient had a surgical procedure for laminectomy and discectomy, and fusion with fixation, fusion without fixation, and joint replacement surgeries were recorded if the patients had a surgical procedure for THR and TKR within 1 year after the index date.

### 2.4. Covariates

Differences in spine reoperation rates among surgical groups may be partly due to differences in patient characteristics. A surgeon may choose to perform fusion with fixation on more difficult and complex patients. We therefore considered patient’s age, sex, urbanization (1 is highest and 3 is lowest), social economic status (SES, 1 is highest and 6 is lowest), and previous or coexisting disease conditions to adjust for differences. We used the Charlson comorbidity index (CCI) to adjust for the severity of disease conditions and patients with a specific disease were defined if the patients had a least two diagnostic claims and the two claims were 4 weeks apart within one year prior to the index date. Medication uses were also taken into consideration and defined if the patient received pharmacy claims more than 3 months within six months before the index date.

### 2.5. Statistical Analysis

The Chi-square test was used to compare the baseline difference among the three surgical groups. The incidence of spine reoperation and joint replacement surgery was estimated by the Kaplan–Meier method. The Cox proportional hazard regression was used to examine the risk factors associated with spine reoperation. Patients who died during the 1-year follow-up period were treated as censored cases. All analyses were performed using SAS/STAT 9.4 software (SAS Institute Inc., Cary, NC) and STATA 14 software (Stata Corp LP, College Station, TX). A *P* value of 0.05 was considered significant.

## 3. Results

### 3.1. Baseline Characteristics

A total of 90,105 patients were identified as having received lumbar discectomy surgery (29,719 patients in fusion with fixation group (33.0%), 2897 patients in fusion without fixation group (3.2%), and 57,489 patients in non-fusion group (63.8%)). The two leading primary diagnoses were disc herniations (64.1%) and spondylolisthesis (15.1%). A relatively large percentage of patients were males (56.8%) and major comorbidities were osteoarthritis (OA) (25.0%), diabetes mellitus (DM) (13.4%), hypertension (HTN) (23.1%), and mechanical insults to the joint from acute injury or repeated loading (20.5%). In the spine fusion with fixation group, there were more patients who had spondylolisthesis, were females ≥ 65 years old, had a CCI ≥ 3, and had a history of oral steroid use. In the non-fusion group, there were more patients who had disc herniation, were males 18–44 years old, and had a CCI = 0 (see Table 1).

### 3.2. Incidence of Spine Reoperation and Joint Replacement

The incidences for 1-year spine reoperation and joint replacement surgery for hip and knee were 0.27, 0.04, and 0.04 per 100 people/month (Table 2). Compared to the fusion with fixation group, the fusion without fixation and non-fusion groups had higher risks of spine reoperation, with an adjusted hazard ratio (HR): 2.30, 95% CI: 1.76–3.02 for the fusion without fixation group and aHR: 4.12, 95% CI: 3.57–4.76 for the non-fusion group (Figure 2a). The risk of joint replacement surgery (Figure 2b,c) was low and we did not observe a difference of the three treatment groups when compared the after-adjusted baseline characteristics.

### 3.3. Risk Factors Associated with 1-Year Spine Reoperation

Table 3 describes the risk factors associated with higher 1-year spine reoperation, including surgical type of fusion without fixation and non-fusion; disc herniation and other diagnoses; age of 45 years old and older; male; DM; CCI = 0; lowest of SES; and a history of systemic steroid use. The variables associated with lower risk include female gender; OA; osteoporosis; HTN; mechanical insults to the joint; and CCI = 1–2.

### 3.4. Risk Factors Associated with 1-Year THR or TKR

Table 4 shows the risk factors for higher 1-year THR including age of 45 years old or greater, osteoporosis, lowest of urbanization and SES, a history of systemic steroid, and NSAID use. The variables that reduce risk included DM, coronary artery disease (CAD), and CCI = 1~2. The risk factors for numerous 1-year TKR include age of 45 years old or older, female gender, rheumatoid arthritis, OA, and NSAID use. The variables associated with a lower risk for 1-year TKR included disc herniation and CCI ≥ 1.

### 3.5. Subgroup Analysis

Figure 3 presents the incidence and risk of 1-year spine reoperation of the three surgical types (treating fusion with fixation as a reference) in different age groups, sexes, CCI scores, and OA, CAD, and DM groups. The subgroup analysis for the spine reoperation showed that fusion with fixation maintained the lowest risk of spine reoperation in almost all subgroups, except patients with OA and CAD.

## 4. Discussion

Fusion and non-fusion surgery are both common procedures for lumbar DDD. Spine reoperation is often related to further degeneration, leading to neurologic compression or spinal instability. This population-based cohort study showed that spine fusion with fixation was associated with a lower risk of 1-year spine reoperation but not joint replacement when compared to fusion without fixation and non-fusion discectomy surgery. Additionally, the results were consistent in many subgroups, including age, sex, CCI, and DM groups; however, this was not the case for patients with OA and CAD.

The effect of fusion on spine reoperation rate has been discussed in the literature extensively. In the study by Vorhies et al., the authors found that the fusion group had a significant lower spine reoperation rate (fusion versus decompression: 5.53% and 6.87%, *p* < 0.001) at the 1-year mark for degenerative lumbar spondylolisthesis; however, the protective effect diminished after 3 years [8]. The same outcome was evaluated in the study by Martin et al. using registry data in the US, wherein the rate of spine reoperation in patients with a herniated disc was 6.4% at 1 year [9]. In another prospective cohort study by Irmola et al. containing 433 patients, the spine reoperation rate following instrumented lumbar spine fusion at two years was found to be 12.5% (95% confidence interval (95% Cl): 9.7−16.0) and at four years 19.3% (95% Cl: 15.6−23.8) [10]. Gerling et al. also reported that the incidence of spine reoperation for patients with degenerative spondylolisthesis was 22% at eight years following surgery [11]. It was difficult to compare the rate of spine reoperation among the surgical methods in different populations, medical settings, and study periods. In our study, the reoperation for spine surgery, namely at a rate of 0.27 (per 100 people/month), was very low compared to previous studies. However, the frequency was similarly low among the three groups, which might be due to the patient’s factors, such as willingness to have spine reoperation and the intervention by other treatments such as acupuncture or rehabilitation programs [9].

### 4.1. Surgical Type and the Risk of Spine Reoperation after Lumbar Discectomy Surgery

In the current study, we found that surgical type was associated with the risk of spine reoperation after adjusting the baseline differences among the three surgical groups. In an early study by Martin et al., the authors found a higher proportion of fusion procedures and the introduction of new spinal implants between 1993 and 1997 did not reduce spine reoperation rates [12], although spinal fusion was considered to be associated with wider surgical exposure, more extensive dissection, and longer operative times than lumbar surgery without fusion in the past. Malter et al. found that spinal fusion operations were associated with higher mortality rates compared to laminectomy or discectomy alone, and spine reoperation rates were not lower [13]. However, in a more recent nationwide cohort study by Kim et al., the authors reported that the spine reoperation rates were 18.6%, 14.7%, 13.8%, 12.4%, and 11.8% after laminectomy, nucleolysis, open discectomy, endoscopic discectomy, and fusion, respectively. Although they found no statistical difference between the fusion and non-fusion surgery groups, the spine reoperation rate of fusion was the lowest among various surgical methods, which is similar to our findings. In our study, fusion with fixation had the lowest 1-year spine reoperation rate, which is different from past studies. However, many patients still could have non-fusion surgery as their first option due to the low spine reoperation rate (0.38 per 100 people/month) in our study. In high spine reoperation-risk patients, fusion with fixation seems to be a better option, which provides the lowest spine reoperation rate.

We proposed that the stability of the surgical segment could be an important factor causing the fusion with fixation group to have a lesser spine reoperation rate compared with other groups. The fusion without fixation group had a higher spine reoperation rate compared to the fusion with fixation group, indicating that the fixation increased the stability of the surgical segment. Decompression had the highest spine reoperation of around four times of incidence following fusion with fixation after adjusting for baseline differences. The 1-year reoperation for short-segment spinal surgery usually related to the decompression surgery due to the preservation of the segment motion. Patients with decompressions are at risk for spine reoperation due to restenosis or instability. Therefore, the preservation of segment stability is important to prevent spine reoperation at 1 year.

### 4.2. Patient’s Factors on the Risk of Spine Reoperation after Lumbar Discectomy Surgery

In a national population-based cohort study by Park et al., the authors found that those with disc herniation and spinal stenosis, as well as males have a higher risk for spine reoperation, which is similar to our results [14]. The result is also similar to Piper K et al. who reported on a cohort of 111,892 patients who underwent spinal surgery [15]. Moreover, they reported 20 perioperative factors associated with the risk of spine reoperation. However, Gerling et al. found that spine reoperation was not increased in smokers, those with DM, those who are obese, or those on workman’s compensation, which is not consistent with our findings [11]. Interestingly, all three studies revealed that male gender and the lowest SES were also risk factors for spine reoperation. We think it may be related to the intensity of working, which may be higher in these two populations, that may accelerate the development of second spinal disorders.

### 4.3. Joint Replacement Surgeries after Lumbar Discectomy Surgery

In our study, the joint replacement surgeries were not uncommon at 1 year after spinal surgery (0.08 per 100 people/month). Joint disorder after spine surgery may be a poor prognostic factor for spine surgery. Hip–spine syndrome or knee–spine syndrome are clinically important issues due to the possibility of not recognizing it at initial surgery. With proper guidelines, this situation can be identified and proper steps for treatment can be done [4,16,17]. In this study, we also defined several risk factors for higher 1-year THK and THR. The effect of spinal fusion or decompression seems to not change the rate for joint replacement after spine surgery. If patients are not notified about their joint problem before spine surgery, the second surgery could lead to some legal issues. Therefore, surgeons should evaluate the hip or knee joints before spinal surgeries to determine the surgical strategy best suited to the patients age, sex, and comorbidities.

### 4.4. Strength and Limitations

The strength of our study concerns the large population-based cohort aspect and the completeness of the database. The chance to loose-to-follow-up is very low due to the national cohort basis. This allowed us to identify meaningful risk factors. Additionally, joint surgery after spine surgery has not been measured in the past literature. We found that the type of spine surgery was not associated with the risk of joint surgery. The limitations of our study include a lack of detailed data regarding surgical methods, such as approaches and implants; a lack of demographic data of the patients including their place of living and occupation, use of orthosis, and rehabilitation protocol; and the exclusion of patients who received fusion and non-fusion surgeries without discectomy. Classification of DDD and the medication use only recorded when prescribed within three months before the index date, as presented in the NHI claims data, might lead to information bias. The study only focused on the incidence and risk factors of 1-year spine reoperation and joint replacement surgeries after short-segment spinal surgery; therefore, the effect of different surgical methods on the outcomes of interest in a longer period may be different and requires a further study. Furthermore, the spine reoperation was defined to be second spine surgery without the inclusion of implant removal, wound debridement, or short-term revision. We also recognized that staged operation was not specifically identified from spine reoperations, which might overestimate the incidence. However, for short-segment surgery in DDD, staged operation is not a common approach and therefore we believe the effect on our results should be very minimal.

## 5. Conclusions

Non-fusion surgery and spinal fusion without fixation had a higher risk for 1-year spine reoperation compared to spinal fusion with fixation. Therefore, the higher risk for spine reoperation should be explained to patients who will receive non-fusion surgery, even though the risk is low. Although joint replacement surgeries are not uncommon after spine surgeries and can be a poor prognostic factor after spine surgeries, spine surgeries do not increase the risk for joint replacement surgeries.

## Figures and Tables

**Figure 1 jcm-10-05138-f001:**
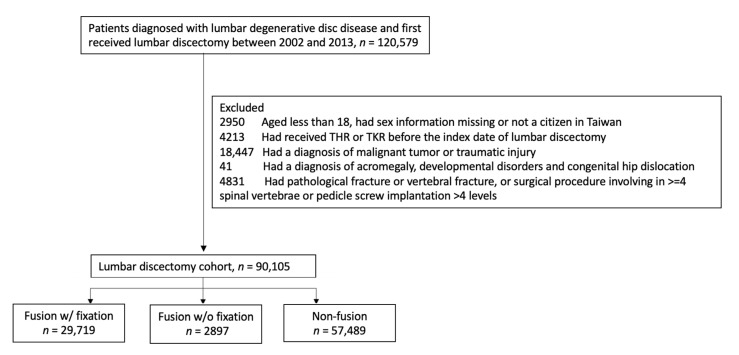
Patient selection process. Abbreviations: THR, total hip replacement and TKR, total knee replacement.

**Figure 2 jcm-10-05138-f002:**
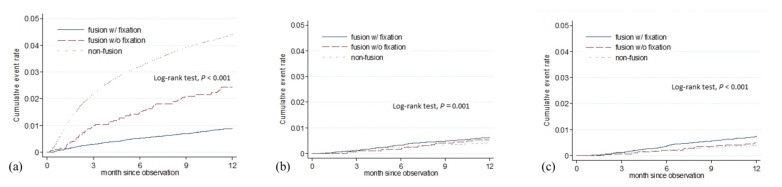
The cumulative event rate of spine reoperation (**a**), total hip replacement (**b**), and total knee replacement (**c**) after short-segment spinal surgery for degenerative disc disease.

**Figure 3 jcm-10-05138-f003:**
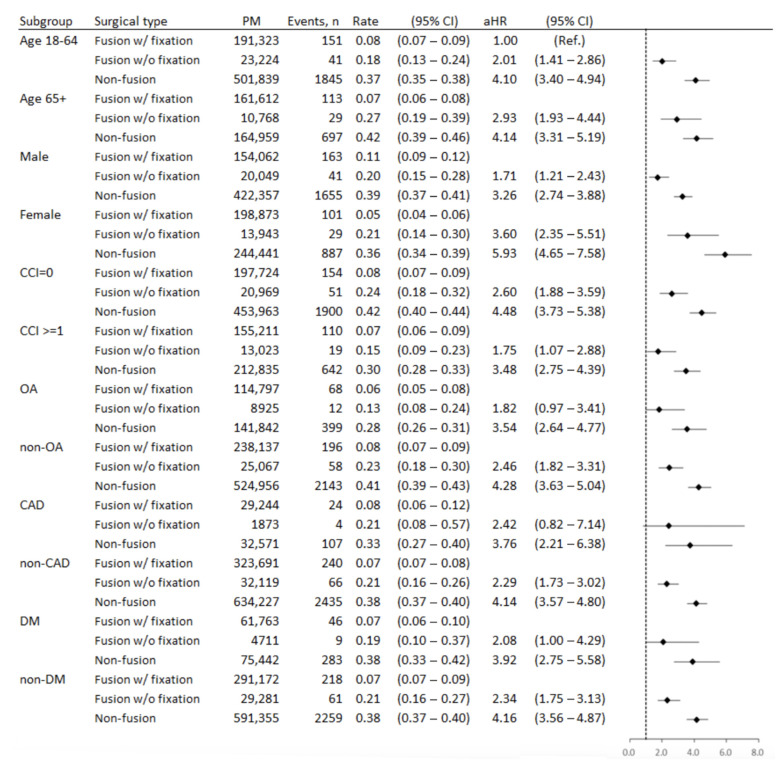
Subgroup analysis of incidence and multivariate Cox proportional hazard ratio of 1-year spine reoperation. Abbreviations: aHR, adjusted hazard ratio; CCI, Charlson comorbidity index; CAD, coronary artery disease; DM, diabetic mellitus; OA, osteoarthritis; and PM, people/month.

**Table 1 jcm-10-05138-t001:** Basic characteristics of patients diagnosed with degenerative disc disease and who first received lumbar discectomy surgery.

	Overall		Fusion with Fixation		Fusion without Fixation		Non-Fusion		
	N	(%)	N	(%)	N	(%)	N	(%)	*P*
Sample size	90,105		29,719		2897		57,489		
Primary disease diagnosis									<0.001
Spondylolisthesis	13,586	(15.1)	12,475	(42.0)	197	(6.8)	914	(1.6)	
Disc herniation	57,792	(64.1)	9738	(32.8)	1800	(62.1)	46,254	(80.5)	
Others	18,727	(20.8)	7506	(25.3)	900	(31.1)	10,321	(18.0)	
Male, yes	51,205	(56.8)	13,028	(43.8)	1711	(59.1)	36,466	(63.4)	<0.001
Age group (y)									<0.001
18–44	29,640	(32.9)	5074	(17.1)	986	(34.0)	23,580	(41.0)	
45–64	31,487	(34.9)	10,986	(37.0)	974	(33.6)	19,527	(34.0)	
65+	28,978	(32.2)	13,659	(46.0)	937	(32.3)	14,382	(25.0)	
Comorbidities, yes									
Rheumatoid arthritis	1134	(1.3)	495	(1.7)	43	(1.5)	596	(1.0)	<0.001
Ankylosing spondylitis	2351	(2.6)	527	(1.8)	86	(3.0)	1738	(3.0)	<0.001
Osteoarthritis	22,484	(25.0)	9641	(32.4)	754	(26.0)	12,089	(21.0)	<0.001
Osteoporosis	3683	(4.1)	1824	(6.1)	120	(4.1)	1739	(3.0)	<0.001
Diabetic mellitus	12,105	(13.4)	5208	(17.5)	405	(14.0)	6492	(11.3)	<0.001
COPD	2983	(3.3)	1209	(4.1)	88	(3.0)	1686	(2.9)	<0.001
Hypertension	20,776	(23.1)	9166	(30.8)	681	(23.5)	10,929	(19.0)	<0.001
Coronary artery disease	5424	(6.0)	2471	(8.3)	162	(5.6)	2791	(4.9)	<0.001
Mechanical insults to the joint from acute injury or repeated loading	18,473	(20.5)	5922	(19.9)	572	(19.7)	11,979	(20.8)	0.004
CCI									<0.001
0	57,739	(64.1)	16,647	(56.0)	1790	(61.8)	39,302	(68.4)	
1–2	26,428	(29.3)	10,479	(35.3)	889	(30.7)	15,060	(26.2)	
3+	5938	(6.6)	2593	(8.7)	218	(7.5)	3127	(5.4)	
Urbanization									<0.001
1 (highest)	51,519	(57.2)	15,827	(53.3)	1700	(58.7)	33,992	(59.1)	
2	31,255	(34.7)	11,043	(37.2)	950	(32.8)	19,262	(33.5)	
3	7331	(8.1)	2849	(9.6)	247	(8.5)	4235	(7.4)	
Social economic status									<0.001
1 (highest)	3648	(4.0)	1185	(4.0)	111	(3.8)	2352	(4.1)	
2	8376	(9.3)	2887	(9.7)	279	(9.6)	5210	(9.1)	
3	38,983	(43.3)	12,410	(41.8)	1232	(42.5)	25,341	(44.1)	
4	31,281	(34.7)	10,967	(36.9)	990	(34.2)	19,324	(33.6)	
5	7139	(7.9)	2070	(7.0)	261	(9.0)	4808	(8.4)	
6	678	(0.8)	200	(0.7)	24	(0.8)	454	(0.8)	
Medication use six months before the index date									
Injection steroid use	13,653	(15.2)	3703	(12.5)	533	(18.4)	9417	(16.4)	<0.001
Oral steroid use	32,142	(35.7)	12,075	(40.6)	1101	(38.0)	18,966	(33.0)	<0.001
NSAID	63,025	(69.9)	21,024	(70.7)	2051	(70.8)	39,950	(69.5)	<0.001

*P* was estimated by the Chi-square test for categorical variables. Abbreviations: CCI, Charlson comorbidity index; COPD, chronic obstructive pulmonary disease; and NSAID, non-steroid anti-inflammatory drug.

**Table 2 jcm-10-05138-t002:** The incidence (per 100 people/month) of spine reoperation and joint replacement surgeries after short-segment spinal discectomy surgeries for degenerative disc disease.

Outcomes	Group	People per Month	Events, *n*	Rate	(95% CI)	Univariate Analysis			Multivariate Analysis		
						HR	(95% CI)	*P*	aHR	(95% CI)	*P*
Spine reoperation	Overall	1,053,724	2876	0.27	(0.26–0.28)						
	Fusion with fixation	352,934	264	0.07	(0.07–0.08)	1.00	(Ref.)		1.00	(Ref.)	
	Fusion without fixation	33,992	70	0.21	(0.16–0.26)	2.74	(2.11–3.57)	<0.001	2.30	(1.76–3.02)	<0.001
	Non-fusion	666,798	2542	0.38	(0.37–0.40)	5.07	(4.47–5.76)	<0.001	4.12	(3.57–4.76)	<0.001
THR	Overall	1,073,150	435	0.04	(0.04–0.04)						
	Fusion with fixation	353,568	186	0.05	(0.05–0.06)	1.00	(Ref.)		1.00	(Ref.)	
	Fusion without fixation	34,394	15	0.04	(0.03–0.07)	0.83	(0.49–1.40)	0.485	0.98	(0.57–1.67)	0.939
	Non-fusion	685,188	234	0.03	(0.03–0.04)	0.65	(0.54–0.79)	<0.001	0.90	(0.71–1.14)	0.379
TKR	Overall	1,073,207	463	0.04	(0.04–0.05)						
	Fusion with fixation	353,423	222	0.06	(0.06–0.07)	1.00	(Ref.)		1.00	(Ref.)	
	Fusion without fixation	34,401	14	0.04	(0.02–0.07)	0.65	(0.38–1.11)	0.115	1.01	(0.58–1.75)	0.971
	Non-fusion	685,383	227	0.03	(0.03–0.04)	0.53	(0.44–0.63)	<0.001	1.13	(0.90–1.41)	0.300

Adjusted for variables listed in Table 1. Abbreviations: aHR, adjusted hazard ratio; CI, confidence interval; THR, total hip replacement; and TKR, total knee replacement.

**Table 3 jcm-10-05138-t003:** Factors associated with the risk of 1-year spine reoperation based on multivariate Cox regression analysis.

Variables	aHR	(95% CI)	*P*
Surgical type (Ref. = fusion with fixation)	1.00	(Ref.)	
Fusion without fixation	2.30	(1.76–3.02)	<0.001
Non-fusion	4.12	(3.57–4.76)	<0.001
Disease dx (Ref. = spondylolisthesis)	1.00	(Ref.)	
Disc herniation	1.80	(1.44–2.24)	<0.001
Others	1.59	(1.27–1.99)	<0.001
Age group (Ref. = 18–44)	1.00	(Ref.)	
45–64	1.68	(1.54–1.85)	<0.001
65+	1.99	(1.79–2.22)	<0.001
Sex (Ref. = male)	1.00	(Ref.)	
Female	0.88	(0.81–0.95)	0.001
Comorbidities, yes			
Rheumatoid arthritis	0.99	(0.66–1.49)	0.961
Ankylosing spondylitis	0.84	(0.66–1.08)	0.172
Osteoarthritis	0.70	(0.64–0.78)	<0.001
Osteoporosis	0.64	(0.49–0.83)	0.001
Diabetic mellitus	1.27	(1.09–1.48)	0.003
COPD	0.78	(0.60–1.02)	0.075
Hypertension	0.70	(0.63–0.79)	<0.001
Coronary artery disease	0.98	(0.82–1.17)	0.811
Mechanical insults to the joint from acute injury or repeated loading	0.63	(0.57–0.70)	<0.001
CCI (Ref. = CCI = 0)	1.00	(Ref.)	
1–2	0.66	(0.59–0.73)	<0.001
3+	0.79	(0.64–0.97)	0.024
Urbanization (Ref. = 1)	1.00	(Ref.)	
2	0.95	(0.88–1.04)	0.273
3	0.98	(0.85–1.14)	0.836
SES (Ref. = 1)	1.00	(Ref.)	
2	1.09	(0.881.36)	0.423
3	1.15	(0.95–1.40)	0.142
4	1.04	(0.85–1.27)	0.685
5	1.11	(0.88–1.39)	0.374
6	2.47	(1.76–3.47)	<0.001
Medication use six months before the index date, yes			
Systemic steroid use	1.39	(1.27–1.52)	<0.001
Oral steroid use	1.08	(0.99–1.17)	0.066
NSAID	0.99	(0.91–1.08)	0.855

Abbreviations: aHR, adjusted hazard ratio; CCI, Charlson comorbidity index; COPD, chronic obstructive pulmonary disease; NSAID, non-steroid anti-inflammatory drug; and Ref, reference.

**Table 4 jcm-10-05138-t004:** Factors associated with the risk of 1-year total hip replacement and total knee replacement based on multivariate Cox regression analysis.

	THR			TKR		
Variables	aHR	(95% CI)	*P*	aHR	(95% CI)	*P*
Surgical type (Ref. = fusion with fixation)						
Fusion without fixation	0.98	(0.57–1.67)	0.939	1.01	(0.58–1.75)	0.971
Non-fusion	0.90	(0.71–1.14)	0.379	1.13	(0.90–1.41)	0.300
Disease dx (Ref. = fpondylolisthesis)						
Disc herniation	0.89	(0.66–1.20)	0.462	0.65	(0.49–0.86)	0.002
Others	1.17	(0.87–1.56)	0.305	1.09	(0.85–1.42)	0.492
Age group (Ref. = 18–44)						
45–64	2.88	(2.06–4.04)	<0.001	4.70	(2.80–7.87)	<0.001
65+	5.21	(3.69–7.35)	<0.001	11.5	(6.91–19.18)	<0.001
Sex (Ref. = male)						
Female	1.02	(0.84–1.25)	0.808	2.19	(1.78–2.70)	<0.001
Comorbidities, yes						
Rheumatoid arthritis	0.78	(0.35–1.78)	0.562	1.93	(1.17–3.18)	0.010
Ankylosing spondylitis	0.89	(0.47–1.67)	0.715	0.89	(0.47–1.67)	0.711
Osteoarthritis	1.09	(0.89–1.35)	0.411	2.36	(1.94–2.88)	0.000
Osteoporosis	1.79	(1.31–2.46)	<0.001	0.91	(0.66–1.26)	0.576
Diabetic mellitus	0.66	(0.46–0.95)	0.024	1.33	(1.00–1.78)	0.051
COPD	1.24	(0.79–1.95)	0.350	1.10	(0.71–1.70)	0.674
Hypertension	0.82	(0.65–1.03)	0.081	0.84	(0.68–1.03)	0.093
Coronary artery disease	0.56	(0.36–0.87)	0.010	1.13	(0.84–1.51)	0.421
Mechanical insults to the joint from acute injury or repeated loading	1.08	(0.86–1.36)	0.490	0.98	(0.78–1.22)	0.848
CCI (Ref. = CCI = 0)						
1–2	0.76	(0.59–0.97)	0.026	0.62	(0.49–0.80)	<0.001
3+	0.88	(0.57–1.38)	0.592	0.64	(0.43–0.95)	0.026
Urbanization (Ref. = 1)						
2	0.93	(0.75–1.16)	0.538	1.14	(0.92–1.41)	0.233
3	1.48	(1.09–2.01)	0.013	1.38	(1.01–1.88)	0.043
SES (Ref. = 1)						
2	1.15	(0.62–2.13)	0.666	1.38	(0.72–2.64)	0.326
3	1.30	(0.75–2.23)	0.352	1.49	(0.83–2.68)	0.181
4	1.36	(0.78–2.37)	0.278	1.53	(0.84–2.77)	0.161
5	1.07	(0.54–2.10)	0.856	1.46	(0.73–2.93)	0.284
6	3.22	(1.34–7.73)	0.009	2.01	(0.64–6.26)	0.230
Medication use six months before the index date						
Systemic steroid use	1.38	(1.09–1.75)	0.007	1.22	(0.96–1.55)	0.106
Oral steroid use	1.16	(0.95–1.41)	0.144	1.08	(0.89–1.30)	0.451
NSAID	1.59	(1.23–2.05)	<0.001	1.43	(1.10–1.86)	0.008

Abbreviations: aHR, adjusted hazard ratio; CCI, Charlson comorbidity index; COPD, chronic obstructive pulmonary disease; NSAID, non-steroid anti-inflammatory drug; Ref., reference; THR, total hip replacement; and TKR, total knee replacement.

## Data Availability

All the study data are reported in this article.

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
