# Peer review of "Risk Factors for Spine Reoperation and Joint Replacement Surgeries after Short-Segment Lumbar Spinal Surgeries for Lumbar Degenerative Disc Disease: A Population-Based Cohort Study"

_jcm, 2021, doi:10.3390/jcm10215138_

Round 1
Reviewer 1 Report
The authors attempt to perform a population-based review of reoperation and joint replacement surgeries after short-segment lumbar spinal surgery. A retrospective analysis of the Taiwan National Health Insurance Research Database was performed. Patients who underwent spine surgery between 2002-2013 were included. 1 year spine reoperation and joint replacement rates were analyzed. A regression analysis was utilized to determine independent predictors of spinal reoperation or joint replacement after index spinal surgery.
90,105 patients were included. Spine reoperation, hip replacement, and knee replacement rates were 0.27, 0.04, and 0.04 per 100-person-month, respectively. The authors found that fusion without fixation and non-fusion groups had higher risk of spine reoperation. Risk factors for spine reoperation included fusion without fixation, non-fusion surgery, age >45, male gender, diabetes, CCI 0, lower socioeconomic class, and history of steroid use. Spine surgery was not found to be a risk for joint replacement surgery.
In the methods section, the authors state that they only included patients who had a lumbar discectomy. However, fusion surgeries and non-fusion lumbar spinal surgeries often do not include discectomies. This should be re-evaluated to include a wider range of operative pathologies.
While the methodologies of the study are sound, the manuscript does not present novel information. In the introduction, the authors state that spine surgery often leads to knee and hip replacement. However, their own data suggests this is not actually the case. The inclusion of total joint replacement rates after spinal surgery within this manuscript holds little value given the low rates of these procedures postoperatively. Why are the authors asking this question? What is the clinical relevance?
The remaining text focuses on risk factors for reoperation after lumbar spinal surgery. This topic has been exhausted in the literature. The authors risk-factors have been well described previously, as they discuss in the discussion section.
Author Response
Comment 1: In the methods section, the authors state that they only included patients who had a lumbar discectomy. However, fusion surgeries and non-fusion lumbar spinal surgeries often do not include discectomies. This should be re-evaluated to include a wider range of operative pathologies.
Reply to comment 1:
Thank you for your suggestion. We only focused on patients who had received discectomy and reoperation rate for each surgical methods including fusion with fixation, fusion without fixation, and decompression with only the index level. Fusion surgeries and non-fusion surgeries without discectomy are not included in this study and we will list it into our limitations.
P10L265 “The limitations of our study are lack of detailed data regarding to surgical methods such as approaches and implants, the demographic data of the patients including the habit and occupation, the use of orthosis, rehabilitation protocol and the exclusion of patients who received fusion and nonfusion surgeries without discectomy.”
Comment 2: While the methodologies of the study are sound, the manuscript does not present novel information. In the introduction, the authors state that spine surgery often leads to knee and hip replacement. However, their own data suggests this is not actually the case. The inclusion of total joint replacement rates after spinal surgery within this manuscript holds little value given the low rates of these procedures postoperatively. Why are the authors asking this question? What is the clinical relevance?
Reply to comment 2: Thank you for pointing out this question. As the literature reported, hip and knee arthritis could occur with spinal pathologies. However, the association in the real-world data haven’t been examined. Therefore, our study also aimed to understand the 2nd surgery due to the concomitant hip or knee pathology and its association with primary spinal surgery. The result showed there’s no association between spinal surgery and 2nd joint replacement surgery, this provides good evidence that hip or knee spine syndrome is not caused by the surgery. We revised our conclusion for better interpretation of its clinical relevance.
P10L281“Although joint replacement surgeries are not uncommon after spine surgeries and can be a poor prognostic factor after spine surgeries, spine surgeries do not increase risk for joint replacement surgeries.”
Reviewer 2 Report
It was an interesting read about this study. thank you.
I have made some comments below.
- Please correct the words (Medical terminology) in the manuscript.
->lumber discectomy, lumbar discectomy
-After short-segment lumbar spinal surgery, the patient's exercise and the presence or absence of orthosis are thought to affect spin reoperation and joint replacement surgeries. Did you have any data on this? If not, it should be included as a limitation of the study.
-In the introduction section, "the study aims to investigate the incidence and risk factors for spine reoperation and joint replacement surgeries including total hip replacement (THR) and total knee replacement (TKR) after short-segment lumbar spinal surgery for DDD." described.
->The conclusion section requires conclusions about joint replacement surgeries.
Thank you.
Author Response
Comment 1: Please correct the words (Medical terminology) in the manuscript.
->lumber discectomy, lumbar discectomy
Reply to comment 1: Thank you for the comment. We have changed “lumber” to “ lumbar” (page 2, line 86 and page 4, line 131) and in figure 1.
Comment 2: After short-segment lumbar spinal surgery, the patient's exercise and the presence or absence of orthosis are thought to affect spine reoperation and joint replacement surgeries. Did you have any data on this? If not, it should be included as a limitation of the study.
Reply to comment 2: Thank you so much for the great comment. We agree this is a limitation due to the nature of the database which did not include information about orthosis. We will add this to the limitations in our study. However, generally almost all patients in Taiwan tend to use orthosis because of the fear of a second surgery which should minimize this bias.
P10L265 ““The limitations of our study are lack of detailed data regarding to surgical methods such as approaches and implants, the demographic data of the patients including the habit and occupation, the use of orthosis, rehabilitation protocol and the exclusion of patients who received fusion and nonfusion surgeries without discectomy.”
Comment 3: In the introduction section, "the study aims to investigate the incidence and risk factors for spine reoperation and joint replacement surgeries including total hip replacement (THR) and total knee replacement (TKR) after short-segment lumbar spinal surgery for DDD." described.
->The conclusion section requires conclusions about joint replacement surgeries.
Reply to comment 3:
Thank you so much and we revised our conclusion to include joint replacement surgeries.
P10L281 “Although joint replacement surgeries are not uncommon after spine surgeries and can be a poor prognostic factor after spine surgeries, spine surgeries do not increase risk for joint replacement surgeries.”
Round 2
Reviewer 1 Report
The changes made to the manuscript are acceptable.